# A Cross-Sectional Study on the Associations between Economic, Social, and Political Resources and Subjective Caregiver Burden among Older Spousal Caregivers in Two Nordic Regions

Sarah Åkerman *, Fredrica Nyqvist and Mikael Nygård

Social Policy, Faculty of Education and Welfare Studies, Åbo Akademi University, 65100 Vaasa, Finland
* Correspondence: sarah.akerman@abo.fi

**Abstract:** Inspired by the caregiver stress process model emphasising the role of resources for caregiving outcomes, the aim of this study was to investigate the prevalence of subjective caregiver burden (SCB) and its associations with individual social, economic, and political resources among older spousal caregivers in a Nordic regional setting. Cross-sectional survey data collected in 2016 in the Bothnia region of Finland and Sweden were used, where 674 spousal caregivers were identified and included in the analyses. The descriptive results showed that about half of the respondents experienced SCB. SCB was more common among Finnish-speaking caregivers. Results from the multivariate logistic regression analysis showed that none of the assessed political resources were significantly associated with SCB when controlling for other variables. Experiencing financial strain was associated with SCB, while personal income was not. Frequent contact with family members was statistically significantly associated with SCB. Future research could use longitudinal data to determine causal relationships, and when data allow, test the full caregiver stress process model to investigate the role of mediating factors in different comparative settings. Accumulated evidence on risk factors for negative outcomes of informal caregiving can contribute to effective screening tools for identifying and supporting vulnerable caregivers, which is becoming increasingly important with the ageing population.

**Keywords:** informal care; community dwelling; ageing; spousal caregivers; subjective caregiver burden; caregiver stress process model; ethnolinguistic; resources

## 1. Introduction

Informal caregivers are the backbone of any social and health care system [1,2] and this is also the case in the Nordic countries of Finland and Sweden where care for older adults is formally a public responsibility as opposed to a family obligation. Informal caregivers are generally defined as persons who provide unpaid care to older and dependent persons with whom they have a social relationship [3]. Although informal caregiving may entail positive experiences [4] with positive or no effects regarding some health aspects [5,6], there is still vast evidence on the negative outcomes on wellbeing and/or health associated with providing intensive informal care. Such negative outcomes may include depression and poorer subjective wellbeing [7–10]. Among informal caregivers providing intensive care in Nordic countries, older co-residing spouses are overrepresented [11,12]. Given the expected increased care needs due to the ageing population in many parts of the world, the life situation of informal caregivers warrants further investigation with attention to individual risk factors that can assist in screening for informal caregivers especially at risk of negative outcomes of caregiving.

According to the caregiver stress process model [13], caregiving can be seen as a stress process departing from a specific context where the caregiver's and care recipient's socioeconomic status and health are examples of important influential factors. The caregiving stress process depends on objective stressors, such as the condition of the care recipient

and the type and amount of care provided. These objective (or primary) stressors are transferred into secondary stressors of subjective strains that may include, for example, family conflicts, constriction of social life, and loss of self and/or sense of mastery. The objective and subjective stressors end in outcomes for the caregiver, such as negative health effects or giving up the caregiving role, but available resources and support mechanisms mediate the pathway between stressors and outcomes. Mediating resources exist on both micro, meso, and macro levels [13,14].

Recent international studies have identified the importance of formal service availability for informal caregiver wellbeing [2,15–17], with Finland and especially Sweden representing some of the most generous public welfare systems. Further, due to the class and gender class difference reducing goals of the Nordic welfare model [18], the role of individual resources can be seen as less important in Finland and Sweden. However, some studies in Finland and Sweden have found that accessing services for older adults may still depend on individual resources [12,19–22]. For example, negotiating with service gatekeepers may be easier for those with more resources than for informal caregivers who possess less resources and experience themselves as less influential [23]. Further, informal caregiving activities and the outcomes on wellbeing have been found to affect Swedish informal caregivers differently depending on educational level [24,25]. These inconsistencies present a need to further explore the role of sociodemographic resources for the wellbeing of informal caregivers in Nordic countries as well.

In a European comparison of people aged 16–79 [15], Finland and Sweden hosted the highest numbers of informal caregivers but the lowest share of intensive caregivers (providing care for more than 11 h). The high numbers of caregivers and low numbers of intensive caregivers are believed to be the result of shared care responsibilities. Indeed, friends and other family members are important mediators of support for informal caregivers as members of the social network may not only provide emotional support to the main caregiver, but also decrease the care intensity by sharing care tasks and assisting in accessing services [2,13,15]. Most Finnish informal caregivers receiving formal support perceive themselves as well supported by family members and relatives [26], and according to a report assessing informal care in the general adult population, 57% of Swedish caregivers receive support from friends and family members [27]. However, older Nordic caregivers have been found to not share care tasks as much as informal caregivers in younger generations [12,27]. This means that older Nordic caregivers could be at higher risk of subjective and objective caregiver burden than caregivers in other age cohorts.

As outlined above, individual levels of economic and social resources are commonly included in research on caregiver wellbeing, although the relationships are not entirely clear when it comes to the Nordic countries officially characterized by universalism. Thus far, political resources have received less attention in the literature on care. To our knowledge, the role of political resources has not been studied in previous research on caregiver wellbeing in Nordic countries. A previous study conducted in Israel found that subjective social status, an indicator that can be seen as sharing similarities with internal political efficacy, was associated with more positive caregiving experiences and lower risk of caregiver burn-out among professional care workers [28]. Another study found that high levels of internal political efficacy was statistically significantly associated to higher levels of wellbeing, life satisfaction, and quality of life among informal caregivers providing care to older adults with dementia in UK [29]. We anticipate that political resources, measured as internal political efficacy and political participation, can play a role in the caregiving experience in a similar manner as social and economic resources.

Our study takes place in two Nordic regions—the northern region of Sweden (Västerbotten) and the western parts of Finland (Swedish-speaking Österbotten and Finnish-speaking Pohjanmaa). In a comparison among Swedish- and Finnish-speaking older adults in this region in Finland, the Swedish-speaking have been found to possess more social resources and to be more frequently engaged in voluntary organizations than their Finnish-speaking peers [30]. Given the important role of voluntary organizations for health

promoting work among informal caregivers in Finland [11] and the role of social support for informal caregiver wellbeing [31,32], it is possible that Swedish-speaking caregivers possess health-promoting resources in comparison to Finnish-speaking caregivers. By using a linguistic rather than a geographical division of Österbotten/Pohjanmaa (further described in Materials and Methods), our study contributes to research on cultural differences in informal care [33,34].

In this study, we explore a particular dimension of caregiver wellbeing, namely subjective caregiver burden. Subjective caregiver burden [35,36] is a state characterized by stress, fatigue, and altered self-esteem caused by the negative effects of caregiving. Subjective caregiver burden may "threaten the physical, psychological, emotional, and functional health of caregivers" [32]. By using the caregiver stress process model [13] as a theoretical framework, this study aimed to explore the associations between individual resources (*mediators of support/individual resources*) and subjective caregiver burden (*subjective strain*), while controlling for care intensity (*objective strain*) and background variables (*mediators of support/individual resources*). The research questions were as follows:

- What is the extent of subjective caregiver burden among older spousal caregivers in the northern parts of Sweden and the western parts of Finland? Are there regional differences?
- What are the associations between individual levels of economic, social, and political resources and subjective caregiver burden?

## 2. Materials and Methods

The study reporting was complied with the "Strengthening the Reporting of Observational studies in Epidemiology" (STROBE) guidelines [37].

The analyses were based on a cross-sectional survey carried out in 2016 as part of a larger inter-regional research project called the Gerontological Regional Database (GERDA [38]. The overall aim of the research project is to map living and health conditions of older adults in the Bothnia region in Sweden (Västerbotten) and Finland (Österbotten/Pohjanmaa and Etelä-Pohjanmaa). In 2016, the questionnaire was sent out to every 66-, 71-, 76-, 81-, and 86-year-old living in the rural areas and in the city of Seinäjoki, Finland, whilst to every second one living in the city of Vaasa, Finland and every third in the city of Umeå and in the city of Skellefteå, Sweden. The Bothnia region in Finland is bilingual with about 52% Swedish-speakers and 48% Finnish-speakers. The Finnish region is, despite belonging to the same geographical region, treated here as two separate regions based on language group affiliations. Swedish-speaking participants were coded as belonging to Österbotten, and those with Finnish as their mother tongue in Pohjanmaa and in Seinäjoki in Etelä-Pohjanmaa were coded as belonging to Pohjanmaa. Questionnaires were sent to 14,805 older adults and 9386 participated, resulting in a total response rate of 63%. The questionnaire was answered by 4375 participants in Västerbotten, Sweden, and by 2296 in Österbotten and 2715 in Pohjanmaa, Finland, resulting in a response rate of 70.8%, 61.7%, and 54.9%, respectively.

In the questionnaire, an informal caregiver was defined as 'a person looking after a family member/ . . . /that due to illness, lowered functional capacity, or another reason needs help and support and therefore does not manage independently in everyday life'. Participants were categorised as caregivers if they chose at least one of the two first answering options (loved one in my household, loved one in another household, I do not give informal care to anyone) in response to the question 'Who do you give informal care to?'. Similar self-reported questions have been used to identify informal caregivers in the European Social Survey [15] and the Swedish "Good Aging in Skåne" survey [9].

Among the caregivers identified for this study in the GERDA survey, 674 spousal caregivers were identified by answering 'spouse' to the question 'Who do you help?'.

### 2.1. Outcome Variable: Subjective Caregiver Burden

The question 'As an informal caregiver, have you been worrying about not going to be able to care for your loved one in a proper way? (yes, no)' was used to measure

subjective caregiver burden. A similar item as the one used in this study was categorised as an environmental question in the Caregiver Burden Scale [36].

### 2.2. Measures of Economic, Social and Political Resources

The variables used for testing economic resources were personal income and perceived ability to make ends meet. Personal income measured monthly income after taxes with five answering options (0–500 euros, 501–1000 euros, 1001–1500 euros, 1501–2000 euros, more than 2000 euros). The variable was dichotomised into '0–1000 euros' and '>1000 euros'. The other economic variable was measured by the question 'Is it possible for you to make ends meet?', with four answering options. This question was dichotomised into 'without difficulty' and 'with difficulty' (with some difficulty, with difficulty, with much difficulty).

Social resources were assessed by measuring contact frequency with other social network members than the spouse. One variable measured contact frequency with family members and relatives, while the other variable measured contact frequency with neighbours and friends. The original question 'How often are you in contact with one/several of the following persons?' had five answering options. Both variables were dichotomised into 'frequent contact' (several times a week) if the respondent had contact with at least one person in the category several times a week. 'Infrequent contact' (several times a month, a few times a year, never, the person does not exist) indicated that the respondent was in contact with someone in the category less often than several times a week.

Political resources were measured by internal political efficacy and political participation. Internal political efficacy was assessed with the statement 'I feel strong and influential in society'. This variable was dichotomised into 'high' (fully agree, partly agree) and 'low' (do not agree). Political activity was assessed by the question 'Have you during the last five years engaged in the following activities: contacted a civil servant or trustee, appealed against a decision launched by authorities, written a letter to the editor/an article in a newspaper/journal, signed a petition, participated in a demonstration, boycotted a product?'. A sum variable was created on the basis of these six items and dichotomised into 'high' (yes, many times; yes, occasionally) and 'low' (no, do not remember).

### 2.3. Control Variables

Socio-demographic variables included age, gender, educational level (less than 10 years, 10 years or more), and rural or urban residence. The caregiver's self-rated health was tested with the following question: 'In general, how would you say your health is?'. The variable was dichotomised into 'good' (excellent, very good, good) and 'poor' (fair, poor). As a rough estimation of the intensity of the care provided, a question on formal support was used as a control variable. The caregiver receiving formal support for her/his caregiving tasks indicates that the care recipient needs help with basic routines on a daily basis. The question targeted to the caregiver was 'Do you receive support from the municipality or another organisation for providing care? (for example respite care, economic compensation, service vouchers, etc.)', with the answering options of 'yes, what kind?' and 'no'. Region was also included as one control variable (Västerbotten, Österbotten, and Pohjanmaa).

### 2.4. Analyses

The distribution (%) of all variables was calculated according to region (Table 1). Contingency tables with Pearson's Chi-square tests were used to analyse the bivariate association between subjective caregiver burden and social, economic, and political resources (Table 2). Logistic regressions were conducted by calculating odds ratios (OR) with 95% confidence intervals (CI) for the likelihood of reporting subjective caregiver burden by economic, social, and political variables and control variables (Table 3). Four models were analysed, and the variables were entered stepwise in the following sequence: (1) economic, social, and political resources; (2) region; (3) sociodemographic variables; (4) self-rated health and formal support. To test robustness of the model, multicollinearity statistics were run. Variance influence factors ranged between 1–1.5.

All statistical analyses were performed in the statistical program IBM SPSS Statistics 27 [39].

*2.5. Ethical Considerations*

The study follows the Guidelines of the Finnish Advisory Board on Research Integrity [40]. The data collection was approved by the Regional Ethical Review Board in Umeå, Sweden 7.10.2021 (2021-04965, 05-084Ö).

**3. Results**

As shown in Table 1, less than half of the spousal caregivers reported subjective caregiver burden in Västerbotten, Sweden (42.8%) and Swedish-speaking Österbotten, Finland (43.7%), while the number was higher in Finnish-speaking Pohjanmaa, Finland (53%).

**Table 1.** Distribution of variables among older spousal caregivers in Västerbotten (Sweden), Swedish-speaking Österbotten (Finland), and Finnish-speaking Pohjanmaa (Finland).

| Variable | Total (n = 674) | Västerbotten (n = 343) | Österbotten (n = 178) | Pohjanmaa (n = 153) |
|---|---|---|---|---|
| Age (n = 673) | | | | |
| 66 | 117 (17.4%) | 56 (16.4%) | 29 (16.3%) | 32 (20.9%) |
| 71 | 179 (26.6%) | 94 (27.5%) | 49 (27.5%) | 36 (23.5%) |
| 76 | 149 (22.1%) | 80 (23.4%) | 40 (22.5%) | 29 (19.0%) |
| 81 | 151 (22.4%) | 72 (21.1%) | 35 (19.7%) | 44 (28.8%) |
| 86 | 77 (11.4%) | 40 (11.7%) | 25 (14.0%) | 12 (7.8%) |
| Gender (n = 674) | | | | |
| Female | 353 (52.4%) | 172 (50.1%) | 92 (51.7%) | 89 (58.2%) |
| Male | 321 (47.6%) | 171 (49.9%) | 86 (48.3%) | 64 (41.8%) |
| Education (n = 668) | | | | |
| Lower secondary | 311 (46.6%) | 181 (53.4%) | 78 (43.8%) | 52 (34.4%) |
| Upper secondary | 357 (53.4%) | 158 (46.6%) | 100 (56.2%) | 99 (65.6%) |
| Residence (n = 656) | | | | |
| Rural | 259 (39.5%) | 123 (37.0%) | 98 (56.0%) | 38 (25.5%) |
| Urban | 397 (60.5%) | 209 (63.0%) | 77 (44.0%) | 111 (74.5%) |
| Personal income (n = 653) | | | | |
| 0–1000 euros | 228 (34.9%) | 131 (39.3%) | 55 (31.8%) | 42 (28.6%) |
| >1000 euros | 425 (65.1%) | 202 (60.7%) | 118 (68.2%) | 105 (71.4%) |
| Ability to make ends meet (n = 655) | | | | |
| Low | 245 (37.4%) | 125 (37.2%) | 66 (38.6%) | 54 (36.5%) |
| High | 410 (62.6%) | 211 (62.8%) | 105 (61.4%) | 94 (63.5%) |
| Contact with family members (n = 663) | | | | |
| Infrequent | 271 (40.9%) | 130 (38.5%) | 69 (39.2%) | 72 (48.3%) |
| Frequent | 392 (59.1%) | 208 (61.5%) | 107 (60.8%) | 77 (51.7%) |
| Contact with friends and neighbours (n = 649) | | | | |
| Infrequent | 377 (58.1%) | 178 (53.3%) | 100 (59.2%) | 99 (67.8%) |
| Frequent | 272 (41.9%) | 156 (46.7%) | 69 (40.8%) | 47 (32.2%) |
| Political participation (n = 653) | | | | |
| Low | 288 (44.1%) | 139 (41.5%) | 65 (38.5%) | 84 (56.4%) |
| High | 365 (55.9%) | 196 (58.5%) | 104 (61.5%) | 65 (43.6%) |
| Internal political efficacy (n = 627) | | | | |
| Low | 246 (39.2%) | 137 (42.3%) | 61 (38.6%) | 48 (33.1%) |
| High | 381 (60.8%) | 187 (57.7%) | 97 (61.4%) | 97 (66.9%) |
| Self-rated health (n = 665) | | | | |
| Poor | 295 (44.4%) | 149 (44.2%) | 69 (39.2%) | 77 (50.7%) |
| Good | 370 (55.6%) | 188 (55.8%) | 107 (60.8%) | 75 (49.3%) |
| Formal support for informal care (n = 661) | | | | |
| No | 516 (78.1%) | 290 (86.6%) | 126 (72.8%) | 100 (65.4%) |
| Yes | 145 (21.9%) | 45 (13.4%) | 47 (27.2%) | 53 (34.6%) |
| Subjective caregiver burden (n = 580) | | | | |
| No | 317 (54.7%) | 174 (57.2%) | 80 (56.3%) | 63 (47.0%) |
| Yes | 263 (45.3%) | 130 (42.8%) | 62 (43.7%) | 71 (53.0%) |

Table 2 shows the bivariate associations between subjective caregiver burden and economic, social, and political resources in the three regions and the total sample. Subjective

caregiver burden was statistically significantly associated (*p* < 0.05) with ability to make ends meet in Västerbotten, Sweden and Swedish-speaking Österbotten, Finland, but not in Finnish-speaking Pohjanmaa, Finland. Statistically significant associations between subjective caregiver burden and contact with family members was found in the total sample and in Finnish-speaking Pohjanmaa. Internal political efficacy was statistically significantly associated with subjective caregiver burden in the total sample and in Västerbotten, Sweden.

**Table 2.** Bivariate association between subjective caregiver burden and economic, social, and political resources among older spousal caregivers in Västerbotten (Sweden), Swedish-speaking Österbotten (Finland), and Finnish-speaking Pohjanmaa (Finland), respectively.

| | Total, (n = 263) % | *p* | Västerbotten, Sweden (n = 130) % | *p* | Österbotten, Finland (n = 62) % | *p* | Pohjanmaa, Finland (n = 71) % | *p* |
|---|---|---|---|---|---|---|---|---|
| Personal income | | ns | | ns | | ns | | ns |
| 0–1000 euros | 46.9 | | 46.5 | | 48.9 | | 45.5 | |
| >1000 euros | 44.9 | | 40.2 | | 42.6 | | 56.3 | |
| Ability to make ends meet | | *** | | ** | | * | | ns |
| Low | 56.0 | | 53.2 | | 55.8 | | 63.0 | |
| High | 39.8 | | 37.4 | | 36.8 | | 48.2 | |
| Contact with family members | | ** | | ns | | ns | | ** |
| Infrequent | 38.8 | | 37.6 | | 37.5 | | 42.2 | |
| Frequent | 50.1 | | 45.9 | | 47.7 | | 65.2 | |
| Contact with friends/neighbours | | ns | | ns | | ns | | ns |
| Infrequent | 47.9 | | 45.9 | | 45.5 | | 54.0 | |
| Frequent | 40.4 | | 37.9 | | 40.0 | | 50.0 | |
| Political participation | | ns | | ns | | ns | | ns |
| Low | 45.7 | | 43.0 | | 39.6 | | 54.1 | |
| High | 44.8 | | 42.5 | | 43.7 | | 53.4 | |
| Internal political efficacy | | * | | * | | ns | | ns |
| Low | 50.5 | | 49.6 | | 40.7 | | 65.1 | |
| High | 41.1 | | 35.8 | | 44.0 | | 48.8 | |

* *p* < 0.05, ** *p* < 0.01, *** *p* < 0.001. Ns indicates non-significant result.

In Model 1 (Table 3), poor perceived ability to make ends meet and frequent contact with family members were statistically significantly associated with subjective caregiver burden. In Model 2 (Table 3), where regions were added as control variables, poor perceived ability to make ends meet, frequent contact with family members, living in Pohjanmaa, and low internal political efficacy were statistically significantly associated with subjective caregiver burden. When sociodemographic variables were controlled for in Model 3 (Table 3), poor perceived ability to make ends meet, frequent contact with family members, and living in Finnish-speaking Pohjanmaa, Finland were still statistically significantly associated with subjective caregiver burden. Low internal political efficacy was no longer statistically significantly associated with subjective caregiver burden. In the last Model 4, (Table 3), when we controlled for self-rated health and formal support, poor perceived ability to make ends meet, frequent contact with family members, and living in Finnish-speaking Pohjanmaa, Finland remained statistically significantly associated with subjective caregiver burden. In addition, poor self-rated health and receiving formal support were statistically significantly associated with subjective caregiver burden.

**Table 3.** Odds Ratios (OR) and 95% confidence intervals (CI) for subjective caregiver burden among older spousal caregivers (n = 673) in Västerbotten (Sweden), Swedish-speaking Österbotten (Finland), and Finnish-speaking Pohjanmaa (Finland).

| | Model 1<br>OR (95% CI) | Model 2<br>OR (95% CI) | Model 3<br>OR (95% CI) | Model 4<br>OR (95% CI) |
|---|---|---|---|---|
| Personal income | | | | |
|   >1000 euros | 1.00 | 1.00 | 1.00 | 1.00 |
|   0–1000 euros | 0.81 (0.54–1.21) | 0.86 (0.57–1.29) | 0.80 (0.50–1.27) | 0.77 (0.47–1.25) |
| Ability to make ends meet | | | | |
|   High | 1.00 | 1.00 | 1.00 | 1.00 |
|   Poor | 2.13 (1.45–3.13) *** | 2.12 (1.44–3.13) *** | 2.25 (1.50–3.38) *** | 2.16 (1.43–3.26) *** |
| Contact with family members | | | | |
|   Frequent | 1.00 | 1.00 | 1.00 | 1.00 |
|   Infrequent | 0.59 (1.41–3.13) ** | 0.55 (0.38–0.81) ** | 0.59 (0.38–0.82) ** | 0.55 (0.37–0.81) ** |
| Contact with friends/neighbours | | | | |
|   Frequent | 1.00 | 1.00 | 1.00 | 1.00 |
|   Infrequent | 1.39 (0.96–2.01) | 1.33 (0.92–1.93) | 1.37 (0.93–2.01) | 1.43 (0.97–2.12) |
| Internal political efficacy | | | | |
|   High | 1.00 | 1.00 | 1.00 | 1.00 |
|   Low | 1.44 (1.00–2.07) | 1.50 (1.03–2.16) * | 1.46 (1.00–2.14) | 1.35 (0.91–2.01) |
| Political participation | | | | |
|   High | 1.00 | 1.00 | 1.00 | 1.00 |
|   Low | 1.03 (0.72–1.48) | 0.94 (0.65–1.37) | 0.96 (0.65–1.42) | 0.97 (0.65–1.43) |
| Region | | | | |
|   Västerbotten | | 1.00 | 1.00 | 1.00 |
|   Österbotten | | 1.01 (0.65–1.59) | 0.94 (0.59–1.51) | 0.89 (0.55–1.45) |
|   Pohjanmaa | | 1.91 (1.20–3.03) ** | 2.00 (1.23–3.25) ** | 1.77 (1.08–2.92) * |
| Gender | | | | |
|   Male | | | 1.00 | 1.00 |
|   Female | | | 1.34 (0.90–2.00) | 1.21 (0.80–1.82) |
| Age | | | | |
|   66 | | | 1.00 | 1.00 |
|   71 | | | 1.05 (0.60–1.83) | 0.98 (0.56–1.74) |
|   76 | | | 1.33 (0.74–2.42) | 1.30 (0.71–2.38) |
|   81 | | | 0.71 (0.39–1.31) | 0.67 (0.36–1.24) |
|   86 | | | 1.22 (0.54–2.74) | 0.95 (0.41–2.21) |
| Education | | | | |
|   Higher | | | 1.00 | 1.00 |
|   Lower | | | 0.94 (0.62–1.40) | 0.97 (0.64–1.46) |
| Residence | | | | |
|   Urban | | | 1.00 | 1.00 |
|   Rural | | | 1.26 (0.84–1.88) | 1.24 (0.82–1.88) |
| Self-rated health | | | | |
|   Good | | | | 1.00 |
|   Poor | | | | 1.51 (1.01–2.24) * |
| Formal support for informal care | | | | |
|   No | | | | 1.00 |
|   Yes | | | | 1.70 (1.06–2.74) * |
| −2 Log Likelihood | 676.415 | 668.178 | 645.047 | 627.029 |
| Cox & Snell R Square | 0.059 | 0.074 | 0.092 | 0.106 |
| Nagelkerke R Square | 0.079 | 0.099 | 0.123 | 0.142 |

Model 1 is adjusted for economic, social, and political variables. Model 2 is adjusted for economic, social, political, and regional variables. Model 3 is adjusted for economic, social, political, regional, and sociodemographic variables. Model 4 is adjusted for economic, social, political, regional, sociodemographic, caregiver's self-rated health, and formal support for informal care. * $p < 0.05$, ** $p < 0.01$, *** $p < 0.001$.

## 4. Discussion

This study aimed to study the prevalence of subjective caregiver burden among older spousal caregivers and explore the associations between subjective caregiver burden and individual social, economic, and political resources in a Nordic regional setting. This was

done by using cross-sectional survey data collected among five different older age cohorts in the Bothnia region in Finland and Sweden.

In Västerbotten, Sweden and Swedish-speaking Österbotten, Finland, 8% of the participants were identified as spousal caregivers, while 6% were spousal caregivers in Finnish-speaking Pohjanmaa, Finland (Table 1). Previous research has identified that about one out of six among the Swedish population aged 65 and above are informal caregivers providing care to a close one of any age [27], with the corresponding share in Finland being 12–26% depending on age cohort [41]. Given that most older caregivers in Finland and Sweden provide care for a spouse [24,26,41], the observed prevalence of spousal caregivers in our study (6–8%) could be seen as in line with previous studies.

About half of the spousal caregivers in our study reported experiencing subjective caregiver burden. In Finnish-speaking Pohjanmaa, Finland, it was more common to report subjective caregiver burden (53%) and to receive formal support (35%) than in Swedish-speaking Österbotten, Finland, where 44% reported subjective caregiver burden and 27% received formal support. Formal support for informal care may on one hand indicate alleviating support services offered in-kind and in-cash, but it may also indicate intensive caregiving [11,26]. Previous research has identified lower membership rates in organizations among Finnish-speaking older adults in the Bothnia region [30]. This could tentatively explain the high prevalence of subjective caregiver burden among caregivers in Finnish-speaking Pohjanmaa, Finland, as third sector organizations play a crucial role in supporting informal caregivers in Finland [11] and Sweden [27]. Nonetheless, the observed regional differences in subjective caregiver burden and formal support are issues that warrant further research. In line with the results of a previous study in Sweden where 13% of older caregivers received formal support [9], only 13% of spousal caregivers in our study reported receiving formal support in Västerbotten, Sweden. Formal support has been found to have alleviating effects on subjective caregiver burden [2,16,26], but as mentioned, receiving formal support may also indicate intensive caregiving [24,26] and not all caregivers who receive public services experience that their needs are being adequately or sufficiently met.

Out of the two economic resource indicators, poor perceived ability to make ends meet was highly associated with subjective caregiver burden in all four models while personal income was not. Perceived ability to make ends meet captures the situation of the household and thus is more representative of a dyadic approach [42], which may be deemed especially relevant when investigating a sample of spouses. Still, our finding that personal income was not associated with subjective caregiver burden is contradictory to a previous finding among Swedish caregivers [43]. To validate our findings further, another scaling of income was tested in a re-run of our analysis, but the results remained the same.

According to the caregiver stress process model [13], social resources may serve as important mediators of support for informal caregivers by not only providing emotional support to the main caregiver, but also by decreasing the objective caregiver burden through sharing care tasks and assisting in accessing services [13,16]. Several previous studies investigating the role of social support for subjective caregiver burden have found alleviating effects [2,16,32]. Out of the two social resource indicators assessed in our study, however, only contact with family members was statistically significantly associated with subjective caregiver burden. Frequent contact indicated subjective caregiver burden which could be deemed as frequent contact being a sign of hardship. Both Finland and Sweden represent low levels of familialism norms [15,41], potentially meaning that older caregivers do not ask for help from other family members unless the care intensity is very high. Indeed, older caregivers in Nordic countries have been found to not share care tasks as much as family members in younger generations [12,27]. Nonetheless, our findings warrant further investigation on the causal relationship between subjective caregiver burden and social resources in a Nordic context.

To the best of our knowledge, our study is the first to investigate the role of political resources among caregivers in a Nordic setting. Previous studies conducted elsewhere

have identified associations between a similar indicator, subjective social status, and different aspects of wellbeing among professional care workers in Israel [28] and informal caregivers in UK [29]. The bivariate analysis (Table 2) in our study showed statistically significant associations between internal political efficacy and subjective caregiver burden among spousal caregivers in Västerbotten, Sweden, but this relationship disappeared in the multivariate analyses (model 1–4, Table 3) where no statistically significant associations between subjective caregiver burden and internal political efficacy nor political participation were found.

Guided by previous international research on informal caregiving [8,14,44], our analysis included some common control variables such as gender. In our study, gender was not statistically significantly associated with subjective caregiver burden, which could be interpreted as the Nordic welfare model succeeding in its gender inequality-reducing goals [18]. The gender gap in terms of who becomes an informal caregiver is quite small in the oldest age groups in Finland and Sweden [11,27]. Similarly, other commonly used [8,14] background variables such as educational level, age, and rural or urban residence were not associated with subjective caregiver burden in our study. This could again potentially be attributed to the Nordic welfare model. However, the relationship between subjective caregiver burden and age and educational level could have been better assessed with a dyadic approach including both the caregiver and care recipient.

In our study, the results from the multivariate analysis (model 4, Table 3) showed that poor self-rated health increased the likelihood of reporting subjective caregiver burden. Self-rated health could be seen as an appropriate factor to include when investigating a sample of older caregivers as their own health may be facing greater risks than caregivers in other age groups. Our results thus stress the need for health promoting initiatives for informal caregivers. Nonetheless, it is also possible that subjective caregiver burden causes poorer self-rated health, and future longitudinal studies should explore the causal relationships between the two factors.

*Methodological Limitations and Strengths*

Limitation of the study includes missing details on the objective caregiver burden, such as for example caregiving hours and type of caregiving tasks. According to the stress process model [13], such objective stressors are closely interlinked to subjective stressors, and this relationship has gained support in several studies [8,14]. Therefore, the model used in this study would have been more robust if it had included details on the care recipient's health status and the type and amount of care provided to him or her. Unfortunately, such variables were unavailable in the data. Instead, the caregiver receiving formal support was used as a rough estimation of care intensity. A similar assessment in terms of receiving formal support being equivalent to providing intensive care has been made by other researchers [6,10]. The caregiver stress process model [13] also includes other indicators than the ones included in our analysis, but the entire model was not possible to test due to the data available from the survey aimed for the general adult population.

Subjective caregiver burden is often assessed through multi-item scales [35,36], but due to the data available, we used a single-item question similar to one categorised as an environmental question in the Caregiver Burden Scale by Elmståhl and colleagues [36]. Using a single question limits the ability to investigate different aspects of strain and the validity of the scale, but also has practical advantages as the response rate was high. Single-item questions to determine subjective caregiver burden have also been used in previous studies [9,45].

Financial stress is commonly included as one of the dimensions of subjective caregiver burden [35,36] and can thus be seen as both a dimension of and as an explaining factor for poor wellbeing among informal caregivers. We interpreted individual levels of economic factors as explaining factors for subjective caregiver burden in our study. However, it is also possible that subjective caregiver burden contributes to financial stress, as subjective caregiver burden is likely to be interlinked with a demanding care situation [13], and care

needs usually bring costs [18,19,46]. Future studies investigating the relationship between economic resources and subjective caregiver burden could preferably use longitudinal data to determine causality.

We assessed social resources by measuring contact frequency, while perceived social support has been suggested to be a better measurement tool [32]. Future studies investigating the role of social resources for informal caregivers could use other indicators to more accurately capture the relationship between subjective caregiver burden and social support. In future studies on social support among informal caregivers, other sources of support could be feasible to include, such as, for example, support from social and health care staff, or social and/or peer support received through activities organized by churches, NGOs, and/or municipalities. By including various sources of support, the social context of informal caregivers and its potential effect could be more accurately captured.

One of the strengths of the study includes a comparatively large regional sample of older spousal caregivers (674 respondents), which is not limited to caregivers who receive formal support. The subsample is obtained from survey data collected from a representative sample of older adults. The response rates were high, ranging from 55–71% in the different regions. Still, there is a risk of bias as informants who are healthier may be more willing and able to participate in surveys. Further, not everyone who cares for close ones may identify themselves as caregivers and, therefore, it is possible that not all caregivers were identified in the survey.

Our study contributed to research on subjective caregiver burden by investigating geographical (Sweden) and ethnolinguistic (Finland) regions, but as social and health care services in Finland and Sweden are organized on a municipal (or county) level, future research could include such a perspective.

## 5. Conclusions

Inspired by the caregiver stress process model by Pearlin et al. (1990) emphasising the role of resources for caregiving outcomes, the aim of this study was to investigate the prevalence of subjective caregiver burden and its associations with individual social, economic, and political resources among older spousal caregivers in a Nordic regional setting. Despite comparatively generous public social and health care systems in Finland and Sweden, it was common for spousal caregivers in the Bothnia region to report subjective caregiver burden, especially among Finnish-speaking caregivers in Finland. There was a statistically significant relationship between financial strain and subjective caregiver burden, although no such associations were found with personal income or other sociodemographic variables. Financial strain can be seen as better reflecting the situation of the household than other sociodemographic factors assessing only the caregiver's resources. The bivariate analysis showed a significant relationship between internal political efficacy and subjective caregiver burden, but none of the investigated political resources remained statistically significantly associated with subjective caregiver burden in the multivariate analysis. Results from the multivariate regression analysis further showed that frequent contact with family members was statistically significantly associated with subjective caregiver burden, which could be interpreted as frequent contact being a sign of hardship. This relationship warrants future research, especially with regard to caregivers who lack such resources. The observed ethnolinguistic differences in the prevalence of subjective caregiver burden in Finland also warrant further investigation. Future research on subjective caregiver burden could preferably use longitudinal data to determine causal relationships. Future studies with more data available could also use multilevel analyses to test the full caregiver stress process model and investigate the role of mediating factors in the relationship between objective and subjective caregiver burden and/or other health outcomes in different comparative settings. Accumulated evidence on risk factors for negative outcomes of informal caregiving can assist in developing effective screening tools and support, which is becoming increasingly important with the ageing population.

**Author Contributions:** Conceptualization, S.Å., F.N. and M.N.; methodology, S.Å., F.N. and M.N.; software, S.Å. and F.N.; validation, S.Å., F.N. and M.N.; formal analysis, S.Å., F.N. and M.N.; investigation, S.Å.; data curation, S.Å.; writing—original draft preparation, S.Å.; writing—review and editing, S.Å., F.N. and M.N.; visualization, S.Å.; supervision, F.N. and M.N. All authors have read and agreed to the published version of the manuscript.

**Funding:** The data collection was supported by the Swedish Research Council (grant: K2014-99X-22610-01-6), the Harry Schauman Foundation, the Regional Council of South Ostrobothnia, Svensk-Osterbottniska Samfundet r.f., the Royal Skyttean Society, Vaasa Aktia Foundation, and the Letterstedtska Association. The work by Sarah Åkerman was supported by the Society of Swedish Literature in Finland and Professor Jan-Magnus Jansson's Foundation for Geriatric and Elderly Care Research.

**Institutional Review Board Statement:** The study follows the Guidelines of the Finnish Advisory Board on Research Integrity TENK (https://tenk.fi/sites/tenk.fi/files/HTK_ohje_2012.pdf, accessed on 12 December 2022). The data collection was approved by the Regional Ethical Review Board in Umeå, Sweden 13 October 2016 (2016/367-32, 05-084Ö).

**Informed Consent Statement:** Informed consent was obtained from all subjects involved in the study, by agreeing to participate in the questionnaire after receiving information about the survey.

**Data Availability Statement:** More information about the Gerontological Regional Database (GERDA) is available on the following webpages: http://urn.fi/urn:nbn:fi:att:e917092f-e9c5-4a7b-91b5-bb704e3daeb9 (accessed on 5 December 2022) and https://www.gerdacenter.com/home?setlang=l3 (accessed on 5 December 2022).

**Conflicts of Interest:** The authors declare no conflict of interest.

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
