# Peer review of "A Cross-Sectional Study on the Associations between Economic, Social, and Political Resources and Subjective Caregiver Burden among Older Spousal Caregivers in Two Nordic Regions"

_nursrep, doi:10.3390/nursrep13010034_

Round 1
Reviewer 1 Report
This is a very good manuscript. The study behind the paper is well conducted and relevant. The manuscript is well written and gives the reader a clear picture of what has been done and why. The few limitations have been dealt with accordingly. The authors have done a good job.
Author Response
Thank you very much for your kind feedback on our manuscript.
Reviewer 2 Report
Thank you for this interesting and valuable manuscript. While carer burden has been extensively documented, its relationship with many of the variables you have chosen, in particular the economic and political resources, has not been examined previously to my knowledge.
I have some questions and comments:
Re the following statement on page 8: "Lower resources indicated caregiver burden for every statistically significant variable with one exception: frequent contact with family members indicated caregiver burden". I'm not sure I understand this. Does" lower resources" refer to income or ability to make ends meet? If either of these, it does not show a relationship with other variables except for the geographic districts. If there is no relationship with the frequent contact with family members, suggest this is a separate sentence, rather than separated by a colon (:).
Suggest the Table heading are accompanied by the numbered model they refer to, to make it easier to relate the text and the Tables.
Line 461, after the word "caregiver", the word "burden" should be inserted.
I would note that while there was logical detailed discussion involving availability of services etc in different areas around the main 4 locations, this was all a little bit difficult for an international reader, and may be difficult to understand. Could it be simplified somewhat?
Author Response
Dear reviewer, thank you for your encouraging comments and constructive feedback. Below you can find our response to your comments.
Reviewer comment: Re the following statement on page 8: "Lower resources indicated caregiver burden for every statistically significant variable with one exception: frequent contact with family members indicated caregiver burden". I'm not sure I understand this. Does" lower resources" refer to income or ability to make ends meet? If either of these, it does not show a relationship with other variables except for the geographic districts. If there is no relationship with the frequent contact with family members, suggest this is a separate sentence, rather than separated by a colon (:).
Authors' response: Thank you for noticing that there were errors in the results section relating to the bivariate analysis in Table 2, which naturally caused difficulties for the reader. We have rewritten this section on page 8 and also slightly changed the title of Table 2.
Reviewer's comment: Suggest the Table heading are accompanied by the numbered model they refer to, to make it easier to relate the text and the Tables.
Authors' response: Thank you, we have inserted clearer references to the models and tables throughout the results and discussion.
Reviewer's comment: Line 461, after the word "caregiver", the word "burden" should be inserted.
Author's response: Thank you for noticing this, we updated the manuscript accordingly.
Reviewer's comment: I would note that while there was logical detailed discussion involving availability of services etc in different areas around the main 4 locations, this was all a little bit difficult for an international reader, and may be difficult to understand. Could it be simplified somewhat?
Authors' response: Thank you, we agree that the text was too detailed and decided to delete a few sentences for better reader friendliness.
Reviewer 3 Report
Thank you for an opportunity to read and review this paper.
The authors explore an important and timely topic, which is the increasing need of caregiving. Following Pearlin et al. (1990) model, they examined the prevalence of subjective caregiver burden and its relation with individual social, economic and political resources among older spousal caregivers in the Nordic region. The paper is well written, clearly exposed and the results are well presented. I recommend its publication. Thank you and good luck with your further research.
Author Response
Thank you for your kind and encouraging comments.
Reviewer 4 Report
This paper broadens the understanding of the caregiver stress process model through a wide set of variables related to subjective caregiver burden (SCB). SCB was then investigated in its association with an individual's social, economic and political resources. The paper did very well in highlighting the areas of need for this population of caregivers, though as pointed out there were also area that could be wither highlighted in future research or addressed in the paper in a little more depth including the physical as well as cognitive well-being of the care recipient, the differing types of strain for the caregiver and the fact that contact frequency was important, but only addressed as familial contact. Other types of frequent contact may be at play with other social networks such as caregiver support groups, church groups and clinical and social programming. All in all, this was a very well written and informative paper.
Author Response
Dear reviewer, thank you for kind and encouraging comments. We agree that it would have been feasible to include more details on the care recipient, but unfortunately this was not possible due to the data available. Thank you for pointing out the fact that informal caregivers' social context may include other sources of support than those assessed in our study. We added a few sentences about this in the in discussion section under methodological limitations and strenghts.